# Chronic Lymphocytic Leukemia Progression Diagnosis with Intrinsic Cellular Patterns via Unsupervised Clustering

**DOI:** 10.3390/cancers14102398

**Published:** 2022-05-13

**Authors:** Pingjun Chen, Siba El Hussein, Fuyong Xing, Muhammad Aminu, Aparajith Kannapiran, John D. Hazle, L. Jeffrey Medeiros, Ignacio I. Wistuba, David Jaffray, Joseph D. Khoury, Jia Wu

**Affiliations:** 1Department of Imaging Physics, The University of Texas MD Anderson Cancer Center, Houston, TX 77030, USA; pingjunchen@ieee.org (P.C.); maminu@mdanderson.org (M.A.); jhazle@mdanderson.org (J.D.H.); dajaffray@mdanderson.org (D.J.); 2Department of Pathology, University of Rochester Medical Center, Rochester, NY 14642, USA; elhusseinsiba@gmail.com; 3Department of Hematopathology, The University of Texas MD Anderson Cancer Center, Houston, TX 77030, USA; ljmedeiros@mdanderson.org; 4Department of Biostatistics and Informatics, University of Colorado Anschutz Medical Campus, Aurora, CO 80045, USA; fuyong.xing@cuanschutz.edu; 5Department of Biomedical Engineering, The University of Texas at Austin, Austin, TX 78705, USA; aparajithk@utexas.edu; 6Department of Translational Molecular Pathology, The University of Texas MD Anderson Cancer Center, Houston, TX 77030, USA; iiwistuba@mdanderson.org; 7Department of Technology and Digital Office, The University of Texas MD Anderson Cancer Center, Houston, TX 77030, USA

**Keywords:** chronic lymphocytic leukemia (CLL), accelerated CLL, Richter transformation (RT), large cell transformation, disease progression, cellular feature engineering, unsupervised clustering, feature fusion, feature selection

## Abstract

**Simple Summary:**

Distinguishing between chronic lymphocytic leukemia (CLL), accelerated CLL (aCLL), and full-blown transformation to diffuse large B-cell lymphoma (Richter transformation; RT) has significant clinical implications. Identifying cellular phenotypes via unsupervised clustering provides the most robust analytic performance in analyzing digitized pathology slides. This study serves as a proof of concept that using an unsupervised machine learning scheme can enhance diagnostic accuracy.

**Abstract:**

Identifying the progression of chronic lymphocytic leukemia (CLL) to accelerated CLL (aCLL) or transformation to diffuse large B-cell lymphoma (Richter transformation; RT) has significant clinical implications as it prompts a major change in patient management. However, the differentiation between these disease phases may be challenging in routine practice. Unsupervised learning has gained increased attention because of its substantial potential in data intrinsic pattern discovery. Here, we demonstrate that cellular feature engineering, identifying cellular phenotypes via unsupervised clustering, provides the most robust analytic performance in analyzing digitized pathology slides (accuracy = 0.925, AUC = 0.978) when compared to alternative approaches, such as mixed features, supervised features, unsupervised/mixed/supervised feature fusion and selection, as well as patch-based convolutional neural network (CNN) feature extraction. We further validate the reproducibility and robustness of unsupervised feature extraction via stability and repeated splitting analysis, supporting its utility as a diagnostic aid in identifying CLL patients with histologic evidence of disease progression. The outcome of this study serves as proof of principle using an unsupervised machine learning scheme to enhance the diagnostic accuracy of the heterogeneous histology patterns that pathologists might not easily see.

## 1. Introduction

Chronic lymphocytic leukemia/small lymphocytic lymphoma (CLL) is one of the most common types of leukemia in adults in western countries [1]. Patients with CLL often have an indolent clinical course. However, approximately 10% of CLL patients undergo histologic transformation to aggressive lymphoma, namely diffuse large B-cell lymphoma (DLBCL), also known as Richter transformation (RT). An intermediate stage of progression, designated as accelerated CLL (aCLL), has overlapping clinical and morphologic characteristics of both CLL and RT [2], rendering its diagnosis inordinately challenging, especially in core biopsy specimens. Notwithstanding, providing an accurate diagnosis in patients with features of progressive disease is of paramount importance, as a diagnostic upgrade to a more aggressive disease phase entails a switch to high-intensity chemoimmunotherapy and carries an adverse prognosis.

Over the past decade, artificial intelligence, and especially deep learning algorithms, have been applied widely to digital pathology image analysis, focusing on the identification and classification of tissues, cells, nuclei, and other histologic features [3]. Supervised learning as a mainstay has been leveraged to improve disease diagnosis and has achieved substantial advancements by taking advantage of hand-crafted pathomics features [4], end-to-end deep learning [5,6], or a combination of such approaches [7]. Unsupervised learning is increasingly employed as a knowledge discovery approach to find meaningful intrinsic patterns in data, complementing supervised learning and enhancing biomedical image diagnostic performance [8,9]. However, few studies have explored the potential role of machine learning algorithms in evaluating lymphomas [10,11,12,13,14,15,16,17]. Moreover, given the uncommon incidence of disease transformation in low-grade lymphomas (including CLL), there are limited studies to assess the diagnostic value of AI-based tools in such a context. 

In this study, we are spearheading the application of a data-driven unsupervised method to identify intrinsic cell populations by clustering cells into multiple phenotypes. We identified three phenotypically distinct cell populations, on the basis of which we extracted morphologic and spatial patterns to build a disease progression diagnostic model. We further evaluated feature engineering through unsupervised clustering and compared its performance with other strategies to handle heterogeneous cell populations, including mixed features, supervised features, unsupervised/mixed/supervised feature fusion and selection, as well as convolutional neural network (CNN)-based feature extraction. Furthermore, we performed stability and repeated splitting analysis to validate the reproducibility and robustness of the proposed cellular feature engineering via unsupervised clustering.

## 2. Materials and Methods

### 2.1. Patient Cohort and Data Collection

This study was approved by The University of Texas MD Anderson Cancer Center (MDACC) Institutional Review Board and conducted in accord with the Declaration of Helsinki. We retrospectively retrieved patients diagnosed with CLL, aCLL or RT evaluated at MDACC between 1 February 2009, and 31 July 2021. Inclusion criteria for this study were: (1) availability of lymph node biopsy glass slides; (2) availability of clinical and laboratory data to support the diagnosis in the electronic medical records. The study group included 135 patients on whom 193 biopsy specimens were procured, including 69 slides from 44 CLL patients, 44 slides from 34 aCLL patients, and 80 slides from 57 RT patients. The microscopic diagnosis was confirmed on all cases by two hematopathologists (Joseph D. Khoury and Siba El Hussein), and challenging cases were resolved by a 3rd hematopathologist (L. Jeffrey Medeiros). Of note, a sizable subset (45.5%) of slides from nearly half (49.6%) of the patients in this cohort were obtained and stained in other institutions across the United States, prior to the slides being sent (usually accompanying the patient) to MDACC, which markedly increased the diversity and heterogeneity of the slides.

Glass slides stained with hematoxylin and eosin (H&E) were scanned using Aperio AT2 scanners at an optical resolution of ×20 (0.50 µm/pixel). Scanning was performed in three batches within the same day, using the same scanner and settings. We split the overall study group into training and testing cohorts for the diagnostic model development and validation with a ratio of 1:1. Since this splitting ratio leaves more samples for testing cohorts, compared with the training/testing ratio of 4:1 or 3:1, and is one of the commonly adopted methods [18]. To mitigate random effects and ensure splitting in a balanced manner, we stratified patients by matching gender, age (above or below 60 years), case source (in house prepared vs referred cases), biopsy technique (excisional vs core needle), and overall survival (>40 or <40 months for CLL, >19 or <19 for aCLL, >9 or <9 months for RT) via the propensity-score matching [19]. Overall characteristics of the patients, as well as the training and testing cohorts, are outlined in Table 1.

### 2.2. Region of Interest (ROI) Selection

Since a regular digital slide of a lymph node excisional biopsy specimen with 80,000 × 50,000 pixels could contain more than two million cells, directly analyzing a whole slide image (WSI) at the cellular level is computationally costly. Moreover, the quality of some regions is low and can pose a challenge for automated diagnosis. For example, several tissue artifacts, including folding, crushing, staining artifact, and section thickness can compromise image assessment and result in the deterioration of whole-slide tissue assessments [20]. In addition, many regions in a WSI may not be relevant for diagnosis purposes, including areas with red blood cell extravasation, necrosis, and extensive fibrosis [21,22]. Thus, we adopted the ROI approach to overcome these pre-analytic impediments for cellular feature extraction. This process is inherently like that taken routinely by pathologists when evaluating biopsy specimen materials. We manually selected diagnostically informative ROIs with high tissue quality for subsequent analysis (Figure 1A). The ROIs were selected in a manner that is representative of the final diagnosis, e.g., if a case was finally diagnosed as RT, the ROI was selected to include the predominant large cell population on the WSI and not the small residual CLL foci in the background. To ensure the inclusion of sufficient cells in each ROI, we set the minimum width and height of selected ROIs to 512 pixels, which corresponds to 0.256 mm. The selected number of ROIs in each slide ranged from 1 to 8. Overall, the numbers of selected ROIs in CLL, aCLL, and RT were 159, 141, and 165, respectively, and the average ROIs per slide was 2.4. Reference-free stain normalization with the Vahadane algorithm [23,24] was performed on all ROIs to mitigate color variations across MDACC and outside institutions before subsequent analyses (Figure 1B).

### 2.3. Cell Segmentation and Filtering

We performed nuclear segmentation on selected ROIs using the Hover-Net [25], which is a deep learning model trained on the PanNuke dataset [26] incorporating 19 different tissue types. Additionally, we performed a quantitative evaluation of 15 manually annotated patches of 256 × 256 pixels and achieved an overall mean Dice score of 0.825. The Hover-Net had Dice scores of 0.826 and 0.853 on datasets Kumar [27] and CoNSep [28], respectively. Cell segmentation on our dataset was on par with Hover-Net’s reported performance. A strong agreement between the manual and Hover-Net segmentations was indicated with a Dice score over 0.80, thus laying the foundation for the proposed cellular-based feature engineering. The Hover-Net results were also visually checked by a hematopathologist (SEH) to assure nuclear segmentation quality (Figure 1C).

Nevertheless, realizing that overlapping nuclei post-nuclear segmentation can still pose challenges in downstream analysis, as it may confound nuclear size sensitivity [29], we used the solidity feature to filter out overlapping segmented cells. By empirically tuning the cutoff with an optimal solidity value of 0.84, we observed that most overlapping nuclei were removed. In addition, we focused on segmented nuclei with pixel numbers between 32 and 432, corresponding to 8 µm^2^ and 108 µm^2^, respectively. This range represented a known biologic nuclear size on histologic slides (Figure 1D). Of note, more than 98% of segmented nuclei were retained whereas nuclei of low quality were removed. 

### 2.4. Pathomics Feature Extraction

#### 2.4.1. Newly Proposed Unsupervised Clustering for Cell Phenotyping

Given the prior histologic knowledge and observation of disease progression and transformation consisting of three different lymphoma stages, i.e., CLL progressing to aCLL and finally transforming to RT, it is reasonable to hypothesize that different cell subtypes predominate at different disease stages, which we termed as CLL-like, aCLL-like, and RT-like cells, respectively. Based on this hypothesis, we designed a data-driven unsupervised clustering strategy to identify three intrinsic cell subtypes based on the appearance of nuclei on WSIs (i.e., size and intensity, Figure 1E). In particular, we employed a spectral clustering algorithm for cell subtype phenotyping, since spectral clustering is known to be a robust method to isolate nonconvex and linearly non-separable clusters [30], which is ideal for handling heterogeneous cell populations. However, for patients enrolled in the training set, there were more than 7.5 million cells even after the filtering procedure, rendering it computationally challenging to directly apply our proposed clustering algorithm. To address this issue, we adopted a subsampling scheme, where 5000 cells were randomly sampled from CLL, aCLL, and RT ROIs, respectively, and then clustering was performed on the pooled cell population. Moreover, we repeated the clustering procedure multiple times with different random sets and varying sample sizes to evaluate cell clustering reproducibility. To propagate the clustering labels to the left-out cells, we trained a multiclass logistic regression (LR) model based on cells used for clustering analysis with phenotypes, and then this classifier was applied to label millions of segmented cells into one of the newly discovered phenotypes (Figure 2A). In the end, we extracted two types of features, including cell ratio and density, to characterize individual cell phenotypes. Cell ratio considers the proportions of individual cell phenotypes inside an ROI, and cell density measures the compactness of individual cell types in each ROI. In total, six cellular features were extracted based on the unsupervised clustering scheme and designated as unsupervised features (Table 2). 

#### 2.4.2. Comparison to Alternative Ways of Cell Profiling

In addition to feature extraction via the aforementioned unsupervised clustering approach, we extracted cellular features based on mixed cell populations and small/large cell subtypes defined by supervised learning. Figure 2 illustrates the three different ways of dealing with cells in our study, and Table 2 shows the associated features. In the following, we elaborate on pathomics feature extraction based on mixed cells and supervised learning approaches.

##### Feature Extraction via Mixed Cells

For the mixed cell analysis, no intrinsic subtype distinction was made. Instead, we directly mixed cells to prepare them for feature extraction (Figure 2B), which was used in our previous study [17]. Two characteristic cytologic features (nuclear size and nuclear intensity) and two characteristic spatial distribution patterns (cellular density and cell to nearest-neighbor distance) were extracted for the diagnosis and termed mixed features (Table 2). For nuclear size, nuclear intensity, and cell to nearest-neighbor distance, we measured the value at every single cell and averaged it across all cells inside each ROI. Cellular density was defined as the number of cells inside the ROI divided by the area covered. The details of mixed features extraction were reported in our earlier study [17].

##### Feature Extraction via Supervised Learning

From a pathology practice standpoint, nuclear size is a predominant factor in distinguishing CLL from its accelerated and transformed phases. To capture nuclear size differences, we aimed to define small versus large cell subtypes via a supervised manner (Figure 2C). Based on the training ROIs, we optimized the cutoff value by maximizing the separation margin among CLL, aCLL, and RT. To be specific, by introducing a size cutoff variable, we partitioned all cells into small and large cells in an ROI, where the large cell ratio was defined as the number of large cells over the total number of cells. Then, we calculated the average large cell ratios of three disease subtypes as $R_{\mathrm{CLL}}$, $R_{\mathrm{aCLL}}$, and $R_{\mathrm{RT}}$ with a given cell size cutoff value. An objective function $\left(R_{\mathrm{RT}}-R_{\mathrm{aCLL}}\right)*\left(R_{\mathrm{aCLL}}-R_{\mathrm{CLL}}\right)*\left(R_{\mathrm{RT}}-R_{\mathrm{CLL}}\right)$ was formulated. By optimizing the cutoff value, we maximized the separation among CLL, aCLL, and RT on the training set. By crossing through all possible cell size values from the minimum of 8 µm^2^ to the maximum of 108 µm^2^, we discovered that the value of 24 µm^2^ was the optimal cutoff. Since the ROI disease label information was leaked during the optimal cutoff searching process, we termed the small and large cell isolation process as supervised learning.

Similar to feature extraction of mixed cells, we also extracted cellular features based on nuclear size, intensity, density, and distance. For nuclear size, we measured the large cell ratio. For nuclear intensity, we first generated the probability distribution function (PDF) of the intensity histogram for both small and large cells inside each ROI. We then measured the similarity between small and large cell PDFs with correlation, Chi-Square, and Wasserstein distance, respectively, which resulted in three intensity-related features [31,32]. In addition, we calculated both the small and large cell densities. We also measured four types of cell distance features, including the average small cell to its nearest small cell neighbor distance, small to large, large to small, as well as large to nearest large cell neighbor distance. Overall, ten features were obtained based on cellular phenotyping with supervised learning, which we designated as supervised features (Table 2).

### 2.5. Machine Learning Models for Disease Progression Prediction

#### 2.5.1. Cellular Feature-Based Diagnosis Models

We aimed to build prediction models based on pathomics features to investigate their performance for disease progression diagnosis. In total, we extracted 20 features based on the three different manners to phenotype heterogeneous cells, including six from unsupervised clustering, four from mixed cells, and ten from supervised learning (Table 2). First, we trained separate prediction models for three feature types, as well as the fused pathomics features. We utilized the XGBoost algorithm [33] to perform the diagnosis analysis based upon the extracted features from each ROI. XGBoost is an ensemble classifier and implements gradient boosted decision trees, known to work well on small sample datasets and known to outperform SVM and random forest [34,35]. In addition, we conducted feature selection on the extracted 20 features via the classic meta-transformer algorithm, which employed a tree-based estimator to compute the impurity-based feature importance [36]. The top-ranked features were retained to train an XGBoost model, where we set the threshold at 0.02 to filter less relevant features. In the end, all of the trained models were independently validated in the testing cohort. 

#### 2.5.2. Comparison to the Convolutional Neural Network (CNN)

For comparison purposes, we also employed the patch-based CNN model, which is the most common way for digital slide image analysis since the renaissance of deep learning [37,38,39,40]. Figure 3 illustrates the procedures of using CNN to extract ROI features. We first divided an ROI into non-overlapping patches with 512 × 512 pixels and discarded those patches with less than 90% pixels inside the ROI. We adopted the transfer learning scheme to fine-tune the ResNet-50 model [41], which was pre-trained on the ImageNet [42] on the cropped patches from the training patient cohort. The early stopping mechanism was used to choose the best patch classifier when the training loss began to plateau. With the fine-tuned CNN model, we took the output of the penultimate layer with a dimension of 2048 to extract the patch-wise features. Since an ROI could contain multiple patches, we performed the average pooling on all of its patches’ features. After the patch-based CNN feature extraction, each ROI was represented as a 2048 feature vector and termed as CNN ROI features. To be consistent, we performed the diagnosis on these ROI features with the XGBoost classifier.

#### 2.5.3. Generalizability Evaluation via Repeated Splitting

Furthermore, we evaluated the robustness of the models through repeated splitting analysis, where we randomly split the whole dataset into training and testing cohorts 100 times, stratified at the patient level with a ratio of 1:1. Patient-based splitting was used to avoid selecting ROIs belonging to the same patients in both the training and testing sets, which can lead to information leakage and model overfitting. The XGBoost models were trained specifically on an individual training set and validated on the testing set. We computed the mean and standard deviation of disease diagnosis performance metrics on the 100 times splitting basis to mitigate any potential biases that might be caused by one-time splitting, which further validated our model’s effectiveness.

### 2.6. Statistical Analysis

The accuracy (ACC) and macro-average of the receiver operating characteristic (ROC) area under the curve (AUC) were measured to assess the performance of the disease diagnostic models. The one-tailed *t*-test (upper tail) was employed to measure if the diagnosis performance of one model is statistically better than the other. 

## 3. Results

### 3.1. Discovery and Validation of Three Cellular Subtypes

We identified three cell phenotypes after spectral clustering. CLL-like cells had small nuclear size and low nuclear intensity, aCLL-like cells had small nuclear size but higher nuclear intensity and the RT-like cells had relatively large nuclear size. The multiclass logistic regression (LR) model was fitted with the clustered pseudo-labels and the decision boundaries were obtained to label the remainder of the cells (Figure 4A). With the fitted LR model, we measured the ratios of CLL-like, aCLL-like, and RT-like cells in the CLL, aCLL, and RT slides in both the training and testing cohorts (Figure 4B). We found that the ratios of these three types of clustered cells were quite consistent in training and testing sets across CLL, aCLL, and RT, with the ratio of RT-like cells, gradually increasing from CLL to aCLL, and RT, whereas the ratio of CLL-like cells showed an opposite trend. These results demonstrated cellular phenotype evolution in tandem with CLL progression to RT and confirmed the hypothesis that different cell subtypes are enriched at different disease stages.

### 3.2. Clustering-Based Model Obtains the Best Performance for Disease Progression Prediction

We trained the XGBoost classifier on the ROIs from patients in the training set and evaluated the diagnostic performance in the testing set. Given three manners of pathomic features, the model built on unsupervised features achieved the best diagnostic performance with an accuracy of 0.925 and AUC of 0.978 (Figure 5A). By contrast, the mixed features model (Figure 5B) attained a lower accuracy of 0.849, and the supervised features model had the lowest performance (accuracy = 0.686, AUC = 0.829) (Figure 5C). Next, we combined these three feature types to build a composite model, which attained a modest performance (accuracy = 0.887, AUC = 0.971) between models based on unsupervised and mixed features (Figure 5D). The feature correlation heatmap of these 20 features showed some high and low correlations among certain feature pairs (Figure 6A). For example, the correlation coefficient between the RT-like cell ratio in unsupervised features (feature 3 in Table 2) and the mean cell size in mixed features (feature 7 in Table 2) was 0.939. The redundancy among the fused features as well as some less informative features might have caused this intermediate performance. Further, we applied a feature selection algorithm, which selected 13 features with highly ranked information, including five features from clustering, four from mixed cells, and four from supervised learning (Figure 6B). Interestingly, the top three important features are from the unsupervised feature set, including the CLL-like cell density, CLL-like cell ratio, and aCLL-like cell ratio. By contrast, features from the supervised set are generally associated with lower importance. The model built on these selected features obtained an accuracy of 0.900 and AUC of 0.975 (Figure 5E), outperforming the model that was directly trained on fused features. This result indicates that removing uninformative features can boost the diagnostic performance of a model. Lastly, we evaluated the performance of the deep learning model. Surprisingly, the CNN model had gained moderate performance with an accuracy of 0.795 (Figure 5F), where a considerable number of aCLL patients were misclassified as RT during testing. The AUC of 0.955 achieved by the CNN model still validated its potential. 

### 3.3. Clustering-Based Model Shows High Reproducibility and Robustness

We tested the stochastic effect of a random selection of cells from CLL, aCLL, and RT ROIs on the unsupervised clustering procedure to ensure reproducibility. We report the results of eight experiments of different random cell selections, as shown in Figure 7A. In the first four experiments, where 15,000 cells were randomly selected but with different random states to ensure the stochasticity in the selection, we observed close performance. Accuracy ranged from 0.921 to 0.929 and AUC ranged from 0.977 to 0.979. Next, by changing the number of selected cells, we observed a notable accuracy improvement when the size increased from 9000 (accuracy = 0.912) to 12,000 (accuracy = 0.916) compared with the employed 15,000 (accuracy = 0.925). Performance reached a plateau when we further increased size to 18,000 (accuracy = 0.925) and 21,000 (accuracy = 0.925). 

Next, we conducted repeated training and testing splitting analysis to validate the general performance of the proposed clustering-based cellular feature extraction strategy and others. Here, we randomly split the patients into training and testing cohorts 100 times with a ratio of 1:1, and repeated the diagnostic model construction and validation, where the performance including the accuracy and AUC are shown in Figure 7B. Similar to the single splitting analysis, the proposed unsupervised clustering manner obtained the best mean accuracy and AUC, with the values of 0.902 and 0.973, respectively. The mean accuracies of mixed, supervised, fusion, and selection feature sets were 0.835, 0.662, 0.876, and 0.891, respectively and the mean AUC values of these four compared manners were 0.961, 0.841, 0.969, and 0.973, respectively. Among these approaches, feature selection exhibited the closest outcome with the unsupervised features. Nevertheless, we conducted the one-tailed *t*-test between the unsupervised model and feature selection accuracies and acquired a *p*-value of 0.043, which demonstrates the advantage of the proposed unsupervised cellular feature analysis. In summary, these results demonstrate the superiority and robustness of the proposed unsupervised cellular feature engineering. However, we observed that the mean accuracies of the repeated splitting analysis are generally 1–3% lower compared with the previous training and testing analysis with balanced splitting, suggesting that unbalanced splitting of gender, age, etc. between training and testing cohorts can negatively affect diagnostic performance. 

## 4. Discussion

In this study, we hypothesized that phenotyping cells in CLL and its phases of disease progression and extracting features based on fine-grained cellular phenotypes can enhance diagnostic performance. Subsequently, we discovered and validated three cellular phenotypes with unsupervised clustering, and these cellular subtypes demonstrated the distinct size and intensity features, corroborating with clinical observations of disease transformation in CLL patients [2]. Furthermore, our trained model, based on six pathomics features characterizing these three cellular subtypes, achieved the highest performance for diagnosing the three phases of the disease. By contrast, the alternative ways to extract pathomics features, including mixed and supervised schemes, were associated with lower diagnostic accuracy. Integrated analysis of different types of pathomics features failed to improve prediction accuracy, indicating the non-synergistic interactions among different pathomics feature types. In addition, our proposed unsupervised clustering-based model showed superior performance compared with the state-of-the-art deep learning approach. 

Data suggest that the prognosis of aCLL patients is poorer than that of CLL patients [2], and such patients usually need more intensive treatment regimens [43]. In clinical practice, the distinction between CLL and aCLL in patients being evaluated for progressive disease can be challenging, particularly in core biopsy specimens, the most common sampling biopsy technique for tissue confirmation in CLL patients with features of disease progression. Evaluating such samples usually requires subspecialty expertise, and even with such expertise distinction requires a high degree of proficiency. These challenges make CLL disease phases an ideal substrate for developing deep learning algorithms that can serve as diagnostic aids to address clinical challenges in this space.

Currently, data-driven unsupervised clustering is more widely used in the medical field, for example, in electronic medical records [44,45], medical decision making [46,47], and medical image analysis [8,9,48]. Additionally, several digital pathology studies have adopted unsupervised clustering to represent various features [49,50,51,52]. In an earlier study [17], we showed that a machine learning approach to integrating cell morphologic and spatial patterns achieved some promising results for the diagnosis of CLL progression. However, this previous study treated cells as the same category without distinguishing them, i.e., identical to the mixing feature approach included in this study. Here, this study is among the first to propose an unsupervised learning model to enhance the diagnostic accuracy of CLL progression. Interestingly, three phenotypically distinct cellular populations were identified and validated, and their relative composition changes during disease progression. Furthermore, by taking advantage of both cytologic and spatial cellular features of the newly phenotyped cells, the proposed cellular feature engineering model obtained the best performance among all compared methods, including both conventional pathomics and deep learning methods. The superiority of unsupervised cellular features can also be inferred from feature importance analysis. As shown in Figure 6B, five of six unsupervised features are ranked with high importance, and the least ranked of these five features was greater than the supervised features with the highest importance. Although all four mixed features were retained after feature selection, the values of their importance were generally inferior to the five selected unsupervised features. While these three manners of cell characterization can be seen as profiling tumor microenvironment from different perspectives, direct feature fusion results in a substantial deterioration of model performance, possibly due to the pollution from irrelevant and redundant information. This presented analysis highlights the importance of extracting meaningful pathomics features, which may not follow the rule of the more the better, similar to our observation in radiomics analysis [53]. 

Convolutional Neural Networks [54] (CNNs) are the state-of-the-art in most fundamental image recognition tasks, including image classification [41], object detection [55], and semantic segmentation [56,57]. Hou et al. adapted a patch-based CNN approach for whole slide image (WSI) classification [37], and now this CNN approach has become the off-the-shelf approach for analyzing WSIs, including large ROIs, with applications to cancers in multiple different sites, e.g., bladder [5], thyroid [40], and brain [58]. The main concern in this patch-based manner is the lack of clinical relatability of the extracted CNN deep features, which blends different cell types and their background context through a cascade of the convolutional and pooling operations, similar to classic bulk sequencing. In this study, we took the advantage of the outstanding representation learning power of the CNN algorithm to segment massive nuclei from WSIs. Then, we focused on key components (i.e., different cell types) of high diagnostic value from the heterogeneous histologic landscape and extracted pathomics features accordingly. Different from the patch-based CNN approach, our method can be viewed as a step toward single cell sequencing. Moreover, we emphasized the interpretability of these cellular features, rather than the “black-box” scheme of the classic CNN method. Consequently, the diagnostic performance of our proposed model highlights many advantages of cellular-based features over the CNN extracted features, indicating that a patch-based CNN may not be able to capture subtle cellular transformation during disease progression. Especially, given that aCLL is an intermediate phase between CLL and RT and exhibits overlapping morphologic characteristics of both CLL and RT, the CNN model misclassified a large amount of aCLL patients as RT. This error may have been caused by: (1) a sample of CNN only covers a limited area (512 × 512 pixels), which might hinder its ability to differentiate the three disease entities; and (2) the CNN model may not be well trained with around 100 patient samples rather than millions of new samples in computer vision field since RT is a rare disease. By contrast, the proposed cellular features consider all cells in the given ROI, and this strategy has some advantages by covering a larger tissue field. 

Despite these initial promising results demonstrated by our proposed model, we believe several limitations are worth highlighting. First, this study was a retrospective study with a relatively small cohort (135 patients) from a single hospital. Obtaining a validation cohort from an external institution is a universal challenge for the evaluation of AI models, given the concern of data privacy, as well as the inherent logistic problems on slides digitization and sharing [59]. However, nearly half of the patients were biopsied and slides were stained at other institutions. The proposed study warrants future validation on its generalizability in a prospective multicenter setting. Second, the study was conducted on pre-selected ROIs, instead of WSIs. The cellular-based feature extraction strategy was hindered by the tremendous number of cells within the WSIs, resulting in a heavy computational burden on both nuclear segmentation and cellular feature extractions. By taking advantage of the fast development of high-performance computing techniques, the time cost of nuclei segmentation and feature extraction can potentially be significantly relieved. Third, the proposed cellular feature extraction considered non-cellular regions as background and ignored their potential functionalities in the tumor microenvironment. While the background may have diagnostic implications, it needs to be analyzed in a future study. Lastly, this study proposed a data-driven manner to cluster cells into three phenotypes by factoring in clinical observations. However, the biological underpinnings of these cellular phenotypes are unknown. Limited by the staining technology (H&E), the phenotyping is primarily driven by cell morphology rather than its function, which may not accurately convey the complexity underlying lymphomagenesis [60]. There are exciting advancements in advanced technologies, e.g., multiplex imaging [61,62,63], which can provide high dimensional molecular details for better cellular function phenotyping to improve diagnostic accuracy in the near future.

## 5. Conclusions

In summary, we propose a novel cellular feature engineering via unsupervised clustering to diagnose CLL, aCLL, and RT based on WSIs. Extensive experiments show that the proposed unsupervised features model is superior to mixed features, supervised features, unsupervised/mixed/supervised feature fusion and selection, and CNN features. This study demonstrates the potential for applying an unsupervised clustering approach to enhance the diagnostic accuracy of distinguishing CLL from its progressive phases, aCLL, and RT.

## Figures and Tables

**Figure 1 cancers-14-02398-f001:**
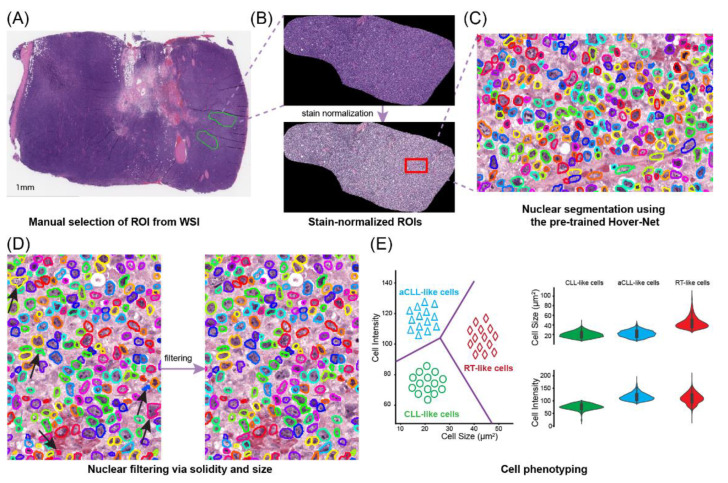
The pipeline cellular feature engineering with unsupervised clustering for diagnosis of chronic lymphocytic leukemia (CLL) progression. (**A**) Manual selection of diagnostically informative ROIs with high tissue quality. (**B**) Stain normalization on ROIs with the reference-free Vahadane algorithm to mitigate undesirable color variations. (**C**) Nuclear segmentation on selected ROIs using the Hover-Net model pre-trained on the PanNuke dataset. (**D**) Filtering segmented nuclei with irregular shape and abnormal size via solidity and size properties. (**E**) Clustering cells into three phenotypes (CLL-like, aCLL-like, and RT-like) with an unsupervised learning method. Differences in cell size and intensity are manifested among these three cell phenotypes.

**Figure 2 cancers-14-02398-f002:**
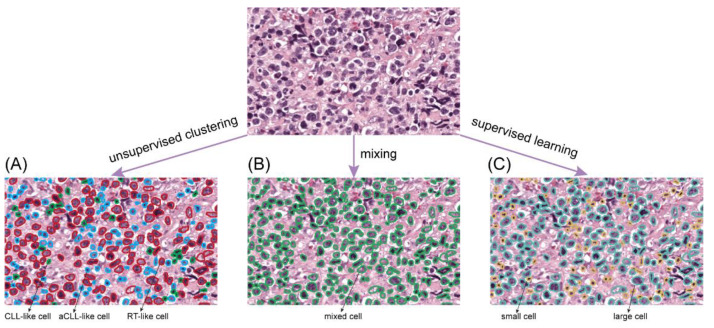
Comparison of three approaches to assess cells: (**A**) Unsupervised learning clusters cells into three categories (CLL-like, aCLL-like, and RT-like). (**B**) Mixing the cells without distinguishing them. (**C**) Supervised learning separates cells into small/large categories.

**Figure 3 cancers-14-02398-f003:**
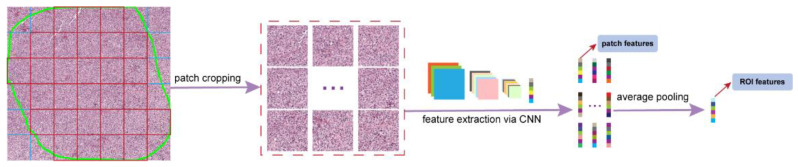
ROI feature extraction with the patch-based convolutional neural networks (CNN). Patches are first cropped from the ROI. The penultimate layer output from the CNN is used to represent each patch. Then average pooling is applied to the extracted patch features to generate the ROI features.

**Figure 4 cancers-14-02398-f004:**
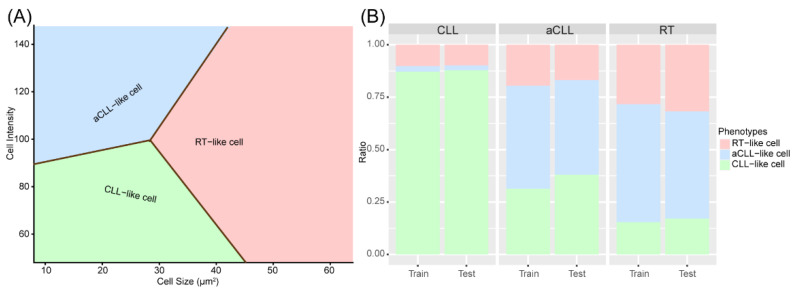
The trained multiple class logistic regression (LR) for labeling CLL-like, aCLL-like, and RT-like cells: (**A**) The decision boundaries of the three cell phenotypes (CLL-like, aCLL-like, and RT-like cells). (**B**) The ratios of the three cell phenotypes in CLL, aCLL, and RT in the training and testing cohorts show consistency.

**Figure 5 cancers-14-02398-f005:**
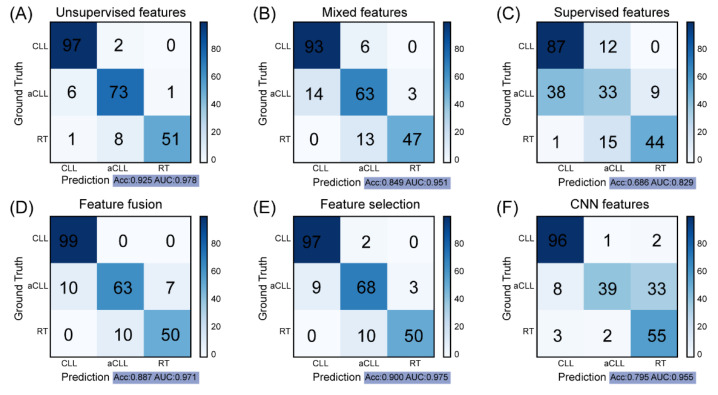
Performance of the proposed unsupervised cellular feature extraction and the five compared methods: (**A**) The proposed supervised features obtain the best performance among all compared methods. (**B**,**C**) presents the diagnostical outcome with the mixed and supervised features, respectively. (**D**,**E**) shows the results based on the feature fusion and feature selection. The feature selection exhibits superior performance compared with mixed, supervised, and fused features. (**F**) presents the performance with the CNN extracted ROI features, which exhibits inferior performance compared with cellular-based feature extraction methods.

**Figure 6 cancers-14-02398-f006:**
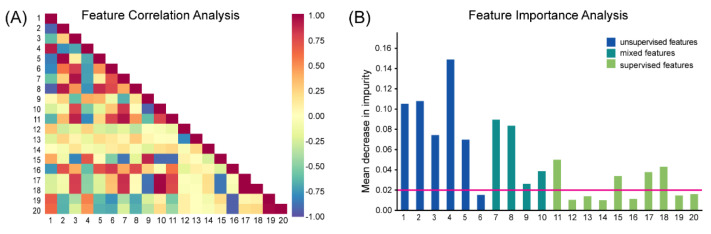
Feature correlation and importance analysis. (**A**) The feature correlation heatmap showed some high and low correlations between certain feature pairs. (**B**) Feature importance analysis based on the tree-based meta-transformer. By choosing 0.02 as the threshold, five in six unsupervised features, all four mixed features, and four in ten supervised features were selected.

**Figure 7 cancers-14-02398-f007:**
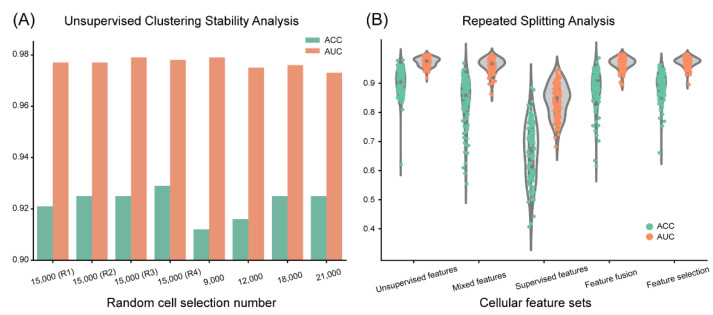
Stability and repeated splitting analysis of the proposed unsupervised clustering cellular feature engineering: (**A**) Eight random cell selections, with four times a random selection of 15,000 cells and one-time respective selection of 9000, 12,000, 18,000, and 21,000 cells, show similar diagnostical performance in general. The accuracy decreases when selecting a smaller number of cells (9000 and 12,000), whereas selecting a larger number of cells (greater or equal to 15,000) presents consistent performance. (**B**) In the 100 times repeated splitting experiment, the extract cellular features via the unsupervised clustering strategy demonstrates superiority over the four compared feature sets on both accuracy and AUC. The feature selection manner exhibits the closest performance with the unsupervised cellular features among all compared methods.

**Table 1 cancers-14-02398-t001:** Training and testing cohorts splitting based on patients’ gender, age, source, biopsy technique, and overall survival.

Attributes		Training Cohort (n = 67)			Testing Cohort (n = 68)		
		CLL (22)	aCLL (17)	RT (28)	CLL (22)	aCLL (17)	RT (29)
Gender	Male	15	11	17	14	13	18
	Female	7	6	11	8	4	11
Age (years)	≥60	9	10	15	9	7	15
	<60	13	7	13	13	10	14
Source	In	11	9	19	6	8	20
	Outside	11	8	9	16	9	9
Biopsy technique	EB	11	7	8	12	7	10
	CNB	11	10	20	9	10	19
OS (M)	Longer	9	7	12	9	7	13
	Shorter	13	10	16	13	10	16

Abbreviations: ≥60: age of CLL diagnosis older or equal to 60; <60: age of CLL diagnosis younger than 60; In: in house case; Outside: outside institution case; EB: excisional biopsy; CNB: core needle biopsy; OS: overall survival; M: month; Longer: CLL ≥ 40|Acll ≥ 19|RT ≥ 9; Shorter: CLL < 40|aCLL < 19|RT < 9.

**Table 2 cancers-14-02398-t002:** Cellular features extracted from unsupervised clustering, mixing, and supervised learning, and termed unsupervised features, mixed features, and supervised features, respectively.

Unsupervised Features	Mixed Features	Supervised Features
1. CLL-like cell ratio	7. mean cell size	11. large cell ratio
2. aCLL-like cell ratio	8. mean cell intensity	12. S/L cell intensity correlation
3. RT-like cell ratio	9. mean cell distance	13. S/L cell intensity chi-square
4. CLL-like cell density	10. cell density	14. S/L cell intensity Wasserstein distance
5. aCLL-like cell density		15. small cell density
6. RT-like cell density		16. large cell density
		17. mean small to small cell distance
		18. mean small to large cell distance
		19. mean large to small cell distance
		20. mean large to large clel distance

Abbreviations: S/L: small/large.

## Data Availability

All the original data of this study were available upon reasonable request to the corresponding author (jwu11@mdanderson.org), including, but not limited to, the request for reproducing the results in this manuscript.

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
