# Peer review of "Chronic Lymphocytic Leukemia Progression Diagnosis with Intrinsic Cellular Patterns via Unsupervised Clustering"

_cancers, 2022, doi:10.3390/cancers14102398_

Round 1

Reviewer 1 Report

The investigated topic is of interest, but several methodological issues must be clarified.

Lines 71-75: the method or a reference for selecting the cutoffs for patient stratification must be included.

Figure 5 shows the results using 239 ROIs from the testing cohort. The text should better clarify the strategy for ROI selection. 

Lines 199-202: A reference for the threshold at 0.02 of the cutoff impurity should be included.

Table 1: "Age" should be changed to "Age (years)."

A reference for the subdivision of the training and test dataset using a 50% and 50% fraction should be provided.

Reviewer 2 Report

Authors declared detection of cellular pattern of CLL by clustering. There are some significance in this study, however, there seems to be critical points to be improved.

  1. The accuracy of cellular segmentation seems to be poor for CLL because Figure 1 shows both the nucleus which could not be segmented correctly and segmented structures other than nucleus including mitosis. The segmentation method should be improved because this procedure is extremely important for this study.
  2. This study does not use the validation cohort from external institution. For confirming validation of this study, the analyses using by cohort of external institution must be adopted.

Reviewer 3 Report

The manuscript is well written and a delight to read. The manuscript can be further consideration for publicaiton after the following concerns are addressed:

(a) " The outcome of this study serves as proof of principle to duly warrant future biomarker studies using an unsupervised machine learning scheme to enhance the diagnostic accuracy of the heterogeneous histology patterns that pathologists might not easily see...." - i think this is over-claiming. I think it is a good proof of principles but to say warrant further biomarker studies is a bit of a stretch to me.

(b) I think we need a table to summarize/compare the results across the different methods. In particular, i am wondering if the authors can do a detailed literature review to see if similar problem statement has been investigated by other groups so that we can do a proper comparison. 

(c) Good that the authors were upfront in stating the limitations of their study. I think they should go one more step in explain how they intend to overcome these limitations as part of their future work.

Round 2

Reviewer 2 Report

None.

Author Response

Thanks for your constructive comments and suggestions.

Reviewer 3 Report

I notice a new section “simple summary”. 
just to confirm that it is a requirement by cancers ?

after that , the revised manuscript Can be accepted. 

Author Response

Thanks for your constructive comments and suggestions. The “simple summary” is a required section for manuscripts submitted to Cancers. We added this part upon the editor's suggestion.